# Poly (Ethylene Oxide)-Based Block Copolymer Electrolytes Formed via Ligand-Free Iron-Mediated Atom Transfer Radical Polymerization

**DOI:** 10.3390/polym12040763

**Published:** 2020-04-01

**Authors:** Sibo Li, Mengying Tian, Jirong Wang, Feipeng Du, Liang Li, Zhigang Xue

**Affiliations:** 1School of Materials Science and Engineering, Wuhan Institute of Technology, Wuhan 430074, China; Sibo_Li@126.com (S.L.); 18040515600@163.com (M.T.); hsdfp@163.com (F.D.); 2Key Laboratory for Material Chemistry of Energy Conversion and Storage, Ministry of Education, School of Chemistry and Chemical Engineering, Huazhong University of Science and Technology, Wuhan 430074, China; wjrhuster@126.com

**Keywords:** macroinitiator, atom transfer radical polymerization, ligand-free, PEO, polymer electrolyte

## Abstract

The Br-terminated poly (ethylene oxide) (PEO-Br) is used as a green and efficient macroinitiator in bulk Fe-catalyzed atom transfer radical polymerization (ATRP) without the addition of any organic ligands. The polymerization rate is able to be mediated by PEO-Br with various molecular weights, and the decrease in redox potential of FeBr_2_ in cyclic voltammetry (CV) curves indicates that an increased coordination effect is deteriorated with the depressing reaction activity in the longer ethylene oxide (EO) chain in PEO-Br. In combination with the study of different catalysts and catalytic contents, the methyl metharylate (MMA) or poly (ethylene glycol) monomethacrylate (PEGMA) was successfully polymerized with PEO-Br as an initiator. This copolymer obtained from PEGMA polymerization can be further employed as a polymer matrix to form the polymer electrolyte (PE). The higher ionic conductivity of PE was obtained by using a high molecular weight of copolymer.

## 1. Introduction

Atom transfer radical polymerization (ATRP) has been extensively used in the synthesis of different functional polymer materials with predictable molecular weights, abundant topology structures as well as low-molecular-weight distributions [1,2,3,4,5,6,7,8,9,10]. Among all the catalyst species in the ATRP process, iron attracts great interests thanks to its considerable availability, less toxicity and relatively lower cost [11,12,13,14,15,16,17,18]. Numerous organic complexes, such as phosphines or amines [17,19,20,21,22], polar solvents [23,24,25] and deep eutectic solvents (DESs) [26,27,28], are employed in order to fulfill the rapid transition of Fe^II^/Fe^III^ by facilitating the iron dissolution and by tuning the metal redox potential. However, such organic ligands are either relatively expensive or toxic, which is detrimental to the environment and biomedical applications.

Low-molecular-weight polyethylene glycols (PEGs) are one kind of environmentally friendly and promising solvents because of their excellent biocompatibility, low cost, nonvolatility and easy degradability [29,30,31,32,33]. The electron-donating ethylene oxide (EO) group endows PEG the ability to coordinate with metal, and thus forms stable complexes and can be applied in organic synthesis, biphasic catalysis and phase-transfer catalysis [34,35,36,37]. Recently, PEG as both the solvent and the ligand in iron-catalyzed ATRP was reported by several groups [38,39]. Fabio di Lena reported on an iron (II) chloride-catalyzed ATRP process in combination with various PEGs, where the homogeneous, “green” catalytic mixtures promoted the well-controlled ATRP of methyl methacrylate in the absence of any additional ligands or solvents [40]. The effects of the polyether structure, molecular weight and chain-end functionality on the polymerization kinetics were also investigated. Afterward, the use of PEG-400 in a FeCl_3_•6H_2_O-mediated activator generated by electron-transfer ATRP (AGET ATRP) was constructed by Zhu and co-workers [41]. PEGs with various molecular weights showed different coordination ability and catalytic activity with an iron catalyst center.

Except for the ligand, initiator species are also essential elements in iron-mediated ATRP, which provide an initiating radical species via homolytic cleavage of its labile carbon-halogen bond. The most widely used initiator includes halo alkanes, alkyl halides, halo ketones or halo esters [42]. Compared with small molecule initiators, the macroinitiator, such as Br-terminated PEO (PEO-Br), is more convenient for the synthesis of AB-type or ABA-type block copolymers, and the obtained PEO-based materials are promising candidates of polymer hosts in solid-state electrolytes for high-energy-density secondary lithium batteries [43,44,45,46,47,48,49,50,51]. Living anionic polymerization and coordination polymerization are the most widely used methods for synthesizing PEO-based copolymer. However, anionic polymerization methods require rigid low temperature and anhydrous and oxygen-free conditions, as well as extreme purity requirements for monomers, solvents and other reagents [52,53]. Thus ATRP is a welcome polymerization method owing to its mild operating conditions. For instance, bromine-terminated diblock copolymer (PEO-*b*-PS-Br) (PS refers to polystyrene) was synthesized by using PEO-Br as the macroinitiator and then converted into azido-terminated PEO-*b*-PS-N_3_ diblock copolymer [54]. In Charleux’s publication, PEO-Br was also used as a reactive surfactant for miniemulsion AGET ATRP of *n*-butyl acrylate, and stable polymer latexes containing polymers with a low-poly-dispersity index were thus obtained [55]. In all these conventional ATRP systems, no system was implemented with one species that combined both the function of the ligand and the initiator [56,57]. The EO group in PEO-Br can coordinate with the iron catalyst. These intriguing properties prompted us to investigate the possibility of using PEO-Br simultaneously as the initiator and the ligands for iron-mediated ATRP.

Herein, PEO-Br was applied as both the ligand and the initiator for the iron-catalyzed ATRP in the absence of any external initiators and organic ligands (Scheme 1). The polymerization rate and controllability were influenced by PEO-Br with different molecular weights (PEO_750_-Br, PEO_1000_-Br, PEO_2000_-Br and PEO_5000_-Br), and well-controlled polymerizations were realized when catalyzed by FeBr_2_. In addition, block copolymer with the water-soluble monomer PEGMA was obtained and could be further used in a polymer electrolyte.

## 2. Experimental Section

### 2.1. Materials

Methyl methacrylate (MMA, 98%, Sinpharm, Wuhan, China) was passed through a column filled with neutral alumina, dried over calcium hydride (CaH_2_), distilled under reduced pressure and stored in a freezer under argon. Triethylamine (TEA, 99%, Sinpharm, Wuhan, China) was dried over calcium hydride (CaH_2_) and distilled under reduced pressure. Tetrahydrofuran (THF, 99+%, Sinpharm, Wuhan, China) was dried over Na and benzophenone and distilled under reduced pressure. Poly (ethylene glycol) monomethacrylate (PEGMA, *M*_n_ = 300 g mol^−1^) was passed through a column filled with neutral alumina and then stored in a freezer. Iron bromide (FeBr_2_, 98+%, Alfa Aesar, Ward Hill, MA, USA), iron tribromide (FeBr_3_, 98+%, Alfa Aesar, Ward Hill, MA, USA), poly(vinylidene fluoride) (PVDF,99%, Aladdin, Shanghai, China), 2-bromoisobutyryl bromide (99.9%, Adamas, Wuhan, China), polyethylene glycol mono-methyl ether (mPEG, 99.9%, Aladdin, Shanghai, China), polyethylene glycol (PEG, 99%, Aladdin, Shanghai, China), sodium ascorbate (AsAcNa, >99%, TCI, Tokyo, Japan), lithium perchlorate (LiClO_4_, 99%, Aladdin, Shanghai, China), and tetraethyl tetrafluoroborate (Et_4_NBF_4_, Aladdin, Shanghai, China) were used without further purification.

### 2.2. Instrumentation

Cyclic voltammetry (CV): All cyclic voltammetry experiments were conducted in a 25 mL four-neck flask with an Autolab PGSTAT302N potentiostat under an Ar atmosphere. The three-electrode system was composed of a Pt sheet (Alfa Aesar, 99.9% metals basis, estimated geometrical area: ~6 cm^2^, Ward Hill, MA, USA) working electrode (WE), a Pt sheet (2 mm diameter, Aldrich, ≥99.9%) counter electrode (CE) and a Ag|AgCl in saturated KCl reference electrode (RE). Prior to each experiment, the Pt electrode was sonicated in ethanol for 5 min.

The electrochemical test system (Autolab PGSTAT302N, Utrecht, The Netherlands) was run to determine the ionic conductivity with temperatures ranging from 30 to 80 °C at 10 °C intervals. The polymer electrolytes (PEs) films were sandwiched between two stainless steel (SS) electrodes, and SS/PEs/SS cells were held at the specified test temperature at least 30 min before measurement. The equation σ = L/(SR) was employed to calculate the ionic conductivity of the PEs. L, S and R are the thickness of electrolyte film, the contact area between the electrode and the PEs, and the bulk resistance, respectively.

The thermal stability was measured by a thermogravimetric analyzer (TGA, PerkinElmer 4000, Wellsley, MA, USA) from 30 to 800 °C under the nitrogen atmosphere with a scanning rate of 10 °C min^−1^.

Gel permeation chromatography (GPC) was selected to obtain the molecular weights and the dispersity of polymers. In this GPC system, a Waters 515 HPLC Pump and a Waters 2414 refractive index detector using polymer standards service columns (7.5 × 300 mm, 10 μm bead size) were assembled. Polystyrene (*M*_n_ = 376~2,570,000) and THF were used as a standard sample and an eluent (a flow rate of 1 mL min^−1^ at 30 °C), respectively.

### 2.3. Preparation of PEO-Br Macroinitiator and Br-PEO_2000_-Br

mPEG (10 mmol) was dissolved into a reaction bottle with the dry THF as the solvent, then the triethylamine (20 mmol) was added at room temperature and the mixture was stirred at 0 °C. The 2-bromoisobutyryl bromide (10 mmol) was added dropwise to the mixed solution and stirred at room temperature for 24 h. The THF solution was added dropwise to excess of cold diethyl ether by filtration under vacuum. The crude white product was obtained and washed three times. The macroinitiator (PEO-Br) was collected and dried in a vacuum. PEO_750_-Br, PEO_1000_-Br, PEO_2000_-Br and PEO_5000_-Br was synthesized by mPEG with different molecular weights (*M*_w_ = 750, 1000, 2000 and 5000), respectively. ^1^H NMR (CDCl_3_, ppm): 4.10–4.20 (CH_2_O, 2H), 3.2 (CH_3_O, 3H) and 1.7–1.8 (C(CH_3_)_2_Br, 6H). The preparation procedure of Br-PEO_2000_-Br was similar to PEO-Br.

### 2.4. Polymerization Procedures

A typical ATRP procedure with the molar ratio of [MMA]_0_:[PEO_2000_-Br]_0_:[FeBr_2_]_0_ = 200:1:1 in the absence of air was as follows: a mixture was obtained by adding PEO_2000_-Br (0.4046 g) and FeBr_2_ (0.0406 g) to a dried ampule in the glovebox. Degassed THF (1.3 mL) and MMA (4 mL) were then added to the flask through degassed syringes. The mixture was stirred for several minutes, then cooled with liquid nitrogen and cycled three times between a vacuum and argon (Ar) to remove oxygen. The solution was defrosted and stirred at room temperature, then the flask was immersed in a thermostated oil bath at 60 °C. The samples were withdrawn from the reaction flask after the desired polymerization time and removed the unconverted monomer and solvent under reduced pressure for calculating the monomer conversion. Tetrahydrofuran (THF) was used as the solvent to dilute the obtained polymers, and the solution was then filtered through a column filled with neutral aluminum oxide to remove the iron catalyst. The block copolymer obtained from the polymerization of MMA (PEO-*b*-PMMA) solution was then precipitated using an excess of *n*-hexane and then washed with water to remove the residual small molecules. These polymers were dried under a vacuum overnight at 60 °C for GPC characterization.

### 2.5. Chain Extension Using PEO-PMMA-Br as the Macroinitiator

A predetermined quantity of PEO-PMMA-Br (obtained by ATRP of MMA) was added to a dried ampule, then a predetermined quantity of MMA and FeBr_2_ was added. The polymerization molar ratio was [MMA]_0_:[PEO-PMMA-Br]_0_:[FeBr_2_]_0_ = 800:1:1. The rest of the procedure was the same as described above. The chain extension polymerization was carried out under stirring at 60 °C.

### 2.6. Preparation of Polymer Electrolyte Membrane

The polymer electrolyte membranes were prepared according to our previous works [58]. PVDF powder, P(EO-PEGMA) copolymer and LiClO_4_ were dissolved in *N*-methyl-2-pyrrolidone (NMP) (solvent) to form a homogeneous solution by stirring at room temperature for 12 h. Then the mixed solution was poured into a circular mold. These polymer electrolyte membranes were dried under a vacuum for 24 h at 80 °C.

## 3. Results and Discussion

### 3.1. Bulk MMA Polymerization with Different PEO-Br as the Initiator and Ligand

In an iron-mediated ATRP process, both the catalyst complexes and initiators have an effect on the polymerization activity. The coordination ability of PEO-Br depends on the electron-donating EO groups, and the number of EO groups in PEO-Br also exerts an influence on initiation activity [40,59]. PEO has the ability to chelate metals, and it can host several catalytic centers when its chain is sufficiently long. Therefore, the ligand-free iron-mediated ATRP process was built using the PEO-Br as both the ligand and the initiator. The ^1^H NMR spectrum of the synthesized PEO_750_-Br and PEO_2000_-Br is shown in Appendix A. The effects of the PEO-Br with different molecular weight and chain-end functionality on the polymerization kinetics were investigated under comparable experimental conditions. 

The bulk MMA polymerization was implemented with PEO-Br as both the ligand and the initiator, and the polymerization molar ratio was [MMA]_0_:[FeBr_2_]_0_:[PEO_x_-Br]_0_ = 200:1:1, x = 750, 1000, 2000 and 5000, respectively. All polymerizations were with linear first-order plots, and the polymerization rate increased with the molecular weight of PEO-Br (Figure 1a). The relative higher monomer conversion (62.1% after 4.5 h) was obtained with PEO_5000_-Br. This means there was a well-coordinated effect between FeBr_2_ and the EO group of PEO-Br, and endowed an improved reaction activity for the catalyst species. The increasing number of EO groups from PEO_750_-Br to PEO_5000_-Br also had a positive effect on the polymerization rate. Linear-molecular-weights growth with the conversion and low-molecular-weight distributions (*Ð* < 1.30) of the PEO_x_-PMMA copolymer were also obtained (Figure 1b). But for the polymerization processes initiated by PEO_1000_-Br and PEO_5000_-Br, the resultant PEO-PMMA polymer showed deviated molecular weights from theoretical ones, which means a radical termination and low initiation efficiency. In addition, the polymerization with less content of catalyst was also implemented in the same condition, and the ratio was [MMA]_0_:[FeBr_2_]_0_:[PEO_x_-Br]_0_ = 200:0.5:1. The polymerization kinetic in Appendix A was still with linear first-order, and *M*_n_ described in Appendix A were slightly higher than the theoretical ones due to the less FeBr_2_ content.

The coordination effect between FeBr_2_ and PEO-Br was further confirmed by cyclic voltammetry, and the results are recorded in Figure 2. The CVs of FeBr_2_ using different PEO-Br exhibited a quasi-reversible peak couple. The half-wave potentials (*E*_1/2_) of catalyst complexes under specific circumstances can be expressed as *E*_1/2_ = (*E*_pa_ + *E*_pc_)/2 [60]. From the CV curves we can see that when PEO_2000_-Br was used as the additive, the reduction potential of iron complex shifted in a negative direction. And as the molecular weight of PEO-Br increased (PEO_5000_-Br), the more negative reduction potential was obtained. The decrease of reduction potential with PEO-Br added means a coordination effect between FeBr_2_ and PEO-Br, which is in agreement with the observed polymerization rate enhancement shown in Figure 1a. In brief, the polymerization results and the CV curves indicate that PEO-Br can replace the conventional organic ligand and the initiator in the iron-catalyzed ATRP process.

### 3.2. Effect of Different Catalysts in Bulk MMA Polymerization with PEO_2000_-Br as the Initiator and the Ligand

In view of both the polymerization rate and controllability, PEO_2000_-Br was chosen as the initiator for the following study. As shown in Figure 3, the effect of different iron catalysts (FeBr_2_, FeCl_2_ and FeCl_2_•4H_2_O) was studied in the bulk MMA polymerization with PEO_2000_-Br as both the ligand and the initiator. The kinetic exhibited the first-order plots, and the polymerization rate decreased in the order FeBr_2_ > FeCl_2_ > FeCl_2_•4H_2_O. Although the molecular weights in FeCl_2_ and the FeCl_2_•4H_2_O-catalyzed polymerization system were slightly higher than the theoretical values, the values increased linearly with monomer conversion. The relative slower polymerization rate and higher molecular weight with the chlorine salts as the catalyst may be due to the fact that a halogen exchange occurred and it generated less active C–Cl. The RCl bond is stronger than RBr, thus the *K*_ATRP_ is lower than RBr. In addition, the coordination ability between FeBr_2_ and PEO_2000_-Br is different from that between FeCl_2_ or FeCl_2_•4H_2_O and PEO_2000_-Br, thus leading to the different reducing power of the iron complex and deteriorative controllability.

### 3.3. Chain Extension of PMMA

Compared with the conventional radical polymerization method, the crucial advantages of reversible deactivation radical polymerization (RDRP) techniques are that it can synthesize the polymer with active chain-ends, which can be functionalized or reinitiated to afford complex macromolecular structures. In order to confirm the activity of the polymer chain-end obtained in this system, the chain extension experiment was carried out using the resulting polymer as the macroinitiator. The macroinitiator PEO-PMMA-Br (*M*_n,GPC_ = 28,400, *M*_w_/*M*_n_ = 1.22) came from the ATRP with a ratio of MMA:FeBr_2_:PEO_2000_-Br = 200:1:1, 60 °C, and the chain-extended polymer was obtained from the ATRP with a ratio of [MMA]_0_:[PEO-PMMA-Br]_0_:[ FeBr_2_]_0_ = 500:1:1, 60 °C. The GPC trace is shown in Figure 4. A peak shift can be seen from the original macroinitiator to the chain-extended PEO-PMMA with increased molecular weight. The successful chain-extension reaction confirmed the activity of the polymer chain-end and the controlled features of ATRP.

### 3.4. Synthesis of the Triblock Copolymer and Water-Soluble Block Copolymer.

Except for the monobromo-terminated PEO-Br, diBr-terminated Br-PEO_2000_-Br was also synthesized for the iron-mediated ATRP process, and the results are listed in Table 1 (entries 1–3). Triblock copolymer PMMA-PEO-PMMA was synthesized, and the monomer conversion gradually increased over time (10.5 h, 42.1%) while maintaining good controllability.

In order to avoid the sensitivity of FeBr_2_ to water and oxygen, the air-stable FeBr_3_ together with the reducing agent AsAcNa were used to synthesize the block copolymer PEO-*b*-PMMA (Table 1, entry 4) with improved controllability, which makes the system more applicable in industrialization.

In addition, the chain length or the number of EO groups in a PEO-based polymer is the essential factor for the material performance, so the water-soluble block polymer P(EO-PEGMA) was also synthesized using PEO_2000_-Br and Br-PEO_2000_-Br as both the ligand and the initiator (Table 2). The diblock copolymer was poorly controlled with higher molecular weight and dispersity (*M*_n_ = 580,800 g/mol, *Ð* = 1.88) in the absence of solvent (Table 2, entry 4). Maybe it was the high viscosity and self-coordination ability of PEGMA that generated a relatively higher concentration of free radicals, thus accelerating the polymerization rate and the extent of termination. Then THF was used as the solvent to improve the homogeneity of the reaction system, thus reducing the concentration of free radicals and avoiding the unnecessary side reactions (Table 2, entries 1–3). The addition of solvent significantly improved the polymerization controllability with lower-molecular-weight distributions, although the molecular weights of the obtained polymer were still a little bit higher than the theoretical ones. However, the diluted effect of THF still cannot compensate the extra self-coordination ability of PEGMA, so the molecular weights of the obtained polymer were much higher than the theoretical ones due to the high instantaneous radical concentration and low initiation efficiency.

### 3.5. The Obtained P(EO-PEGMA) Copolymer for the Application in Polymer Electrolyte.

The PEO-based polymer can complex with lithium salts to form a polymer electrolyte because of its high dielectric constant, strong Li^+^ solvating ability, high donor number for Li^+^ and chain flexibility for promoting ion transport. As mentioned above, the iron-mediated ATRP with PEO-Br as both the ligand and the initiator can be used to polymerize the water-soluble monomer, such as PEGMA. The obtained P(EO-PEGMA) polymer was used as the polymer electrolyte for the electrochemical performance test. As shown in Appendix A, the decomposition temperature of the obtained PMMA, PMMA-PEO_2000_ and PPEGMA-PEO_2000_ was 356, 306 and 252 °C, respectively. Although the stability of the polymers decreased with the increased and flexible EO chain segment, the obtained PPEGMA-PEO_2000_ was still stable enough for the application in polyelectrolyte. In order to suppress the crystalline of the pure PEO-based polymer electrolyte and to let the ion conduction mainly occur in the amorphous phases, PVDF was blended with the obtained block polymer. Table 3 presents the different compositions between the obtained polymer electrolytes and the PVDF, and that the polymer electrolyte material was doped with LiClO_4_.

Figure 5 presents the temperature-dependence ionic conductivity for the P(EO-PEGMA)/PVDF polymer electrolyte doped with LiClO_4_ (EO/Li^+^ = 15). The effect of different molecular weights of P(EO-PEGMA) polymer was studied (the composition of the polymer electrolyte is shown in Table 3, entries 1–3). The original AC impedance spectra are shown in Appendix A. From the results we can see that the ionic conductivities of the polymer electrolyte increased directly proportional to the molecular weight of the P(EO-PEGMA) due to the improved mobility of the polymer segments with the incremental number of EO groups.

In addition, the proportion effect of P(EO-PEGMA) in P(EO-PEGMA)/PVDF polymer electrolyte was also studied and the result was shown in Figure 6 (The composition of the polymer electrolyte was shown in Table 3, entries 3–5). When a relatively higher ratio of P(EO-PEGMA) was used, the ionic conductivities of polymer electrolyte also increased, which was the similarity with traditional solid polymer electrolyte. [43,45,57] All the results in Figure 5 and Figure 6 indicated that the obtained P(EO-PEGMA) polymer from iron-mediated ATRP system with PEO-Br as both the ligand and initiator can act as a potential candidate for application in polymer electrolyte.

## 4. Conclusions

PEO-Br was employed both as the macroinitiator and the ligand in the ATRP system due to its intelligent coordination ability to metal catalyst, which thus fabricated the ligand-free Fe-catalyzed ATRP system. The coordination effect of PEO-Br with different molecular weights (PEO_750_-Br, PEO_1000_-Br, PEO_2000_-Br, PEO_5000_-Br and Br-PEO_2000_-Br) was confirmed by the polymerization rate and the CV. The polymerization process was well controlled and successfully used in the synthesis of block polymer or *tri*-block polymer. Compared with the synthesis method of traditional block copolymers, this system is more convenient and environmentally friendly and has a lower cost. The block polymer P(EO-PEGMA) obtained from the water-soluble monomer PEGMA was blended with PVDF and can be successfully used as a solid polymer electrolyte.

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
