# Peer review of "Poly (Ethylene Oxide)-Based Block Copolymer Electrolytes Formed via Ligand-Free Iron-Mediated Atom Transfer Radical Polymerization"

_polymers, 2020, doi:10.3390/polym12040763_

Round 1
Reviewer 1 Report
This manuscript describes the ligand-free iron-mediated ATRP of methyl methacrylate and poly(ethylene glycol) monomethylacrylate using PEO-Br as macroinitator. It was also reported that the obtained P(EO-PEGMA)block copolymers were used as solid polymer electrolyte after blending with PVDF. I would recommend for publication in Polymers after suitable revisions. A brief list of suggested revisions as is follows;
1- The following statement on page 9 at line 244 is not true "the obtained polymers are still little bit higher than theoretical ones" because Mn (GPC) values of the obtained polymers (Table 2 entries 1-3) much higher than theoretical values. It means that initiation efficiency of polymerization is very low. The authors must correct this statement and add proper discussion/explanation in text.
2- Figure 4. is fairly represented The authors do have expertise in characterization and will know that comparing curves this way is misleading. The authors should compare the GPC chromatograms on a weight basis, since chain extended Poly(EO-MMA) has a much greater mass than the starting macroinitiator.
3- The characterization of all polymers is rather sparsely done. The authors mainly used NMR and GPC. It would be of interest to perform basic material characterization of the obtained polymers e.g. DSC (to get Tg)
4- Figure S1: I wonder why the authors give H NMR of PEO-Br (2000)? H NMR of PEO-Br(750) would be more informative for -Br end group of macroinitiator.
5- Authors should also give an information about the specification of used GPC columns such as length, pore size etc.
Author Response
Reviewer: 1
Comment:
1- The following statement on page 9 at line 244 is not true "the obtained polymers are still little bit higher than theoretical ones" because Mn (GPC) values of the obtained polymers (Table 2 entries 1-3) much higher than theoretical values. It means that initiation efficiency of polymerization is very low. The authors must correct this statement and add proper discussion/explanation in text.
Reply: Indeed,the controllability of the obtained polymers is rather worse than PMMA when used PEGMA as the monomer due to its high viscosity and self-coordination ability, and the diluted effect of THF is not enough for compensating the high concentration of free radical and extent of termination. Thus, although the addition of solvent significantly decreased the molecular weight distributions, the molecular weight of the obtained polymers is still much higher than the theoretical ones. The revised expression was exhibited in the following sentence.
“however, the diluted effect of THF still cannot compensate the extra self-coordination ability of PEGMA, so the molecular weights of the obtained polymer are much higher than the theoretical ones due to the high instantaneous radical concentration and low initiation efficiency.”
Comment:
2- Figure 4. is fairly represented The authors do have expertise in characterization and will know that comparing curves this way is misleading. The authors should compare the GPC chromatograms on a weight basis, since chain extended Poly(EO-MMA) has a much greater mass than the starting macroinitiator.
Reply: As we described in the manuscript, the chain extension experiment was implemented with the ratio of [MMA]0/[PEO-PMMA-Br]0/[ FeBr2]0 = 500:1:1, and the molecular weight of the obtained polymer (Mn,GPC = 107500, Mw/Mn = 1.09) is much higher than the original macroinitiator PEO-PMMA-Br (Mn,GPC = 28400, Mw/Mn = 1.22). So the elution time of the obtained polymer in GPC chromatogram is less than the macroinitiator. So, we think that comparing the curves in this way is reasonable.
Comment:
3- The characterization of all polymers is rather sparsely done. The authors mainly used NMR and GPC. It would be of interest to perform basic material characterization of the obtained polymers e.g. DSC (to get Tg)
Reply: We fairly appreciate the reviewer’s suggestion. Tg is one of the most important property for polymer electrolyte and has been reported in many literatures, the Tg of PMMA-PPEGMA is about -30 oC ~ -60 oC (Journal of Polymer Science: Part A: Polymer Chemistry, 2007, 45, 5770–5780; Journal of Macromolecular Science, Part A: Pure and Applied Chemistry 2015, 52, 252–259). In addition, the thermal stability of polymer electrolyte is also crucial factor for Li-batteries. So we added the relative TGA test of the material of PMMA, PMMA-PEO2000 and PPEGMA-PEO2000, as shown in Figure S4. The revised expression was exhibited in the following sentence.
“The decomposition temperature of PMMA, PMMA-PEO2000 and PPEGMA-PEO2000 is 356 oC, 306 oC and 252 oC, respectively. Although the stability of the polymers decreased with the increased and flexible EO chain segment, but the obtained PPEGMA-PEO2000 is still stable enough for the application in polymer electrolyte. ”
Figure S4. TGA of PMMA, PMMA-PEO2000 and PPEGMA-PEO2000.
Comment:
4- Figure S1: I wonder why the authors give H NMR of PEO-Br (2000)? H NMR of PEO-Br(750) would be more informative for -Br end group of macroinitiator.
Reply: Thanks for your proposal, we supplemented the 1H NMR of PEO-Br (750) together with PEO-Br (2000) in the Figure S1. Indeed, the signal of –OCH2 (4.30 ppm) is much stronger in PEO-Br (750) than PEO-Br (2000).
PEO750-Br |
Comment:
5- Authors should also give an information about the specification of used GPC columns such as length, pore size etc.
Reply: we revised the whole part of experiments, including the he specification of used GPC columns. The revised expression was exhibited in the following sentence.
“mPEG (10 mmol) was dissolved in to a reaction bottle with the dry THF as the solvent, then the triethylamine (20 mmol) was added at room temperature and the mixture was stirred at 0 °C. The 2-bromoisobutyryl bromide (10 mmol) was added dropwise to the mixed solution and stirred at room temperature for 24 h. The THF solution was added dropwise to excess of cold diethyl ether by filtration under vacuum, the crude white product was obtained and washed for three times. [58]The macroinitiator (PEO-Br) was collected and dried in vacuum. PEO750-Br, PEO1000-Br, PEO2000-Br and PEO5000-Br had been synthesized by mPEG with different molecular weights (Mw = 750, 1000, 2000, and 5000), respectively. 1H NMR (CDCl3, ppm): 4.10–4.20 (CH2O, 2H), 3.2(CH3O, 3H), and 1.7–1.8 (C(CH3)2Br, 6H). The preparation procedure of Br-PEO2000-Br is similar to PEO-Br.”
“A typical ATRP procedure with the molar ratio of [MMA]0:[PEO2000-Br]0:[FeBr2]0=200:1:1 in the absence of air is as follows: a mixture was obtained by adding PEO2000-Br (0.4046 g), FeBr2 (0.0406 g) to a dried ampule in the glovebox. Degassed THF (1.3 mL) and MMA (4 mL) were then added to the flask through degassed syringes. The mixture was stirred for several minutes, then cooled with liquid nitrogen and cycled three times between vacuum and argon (Ar) to remove oxygen. The solution was defrosted and stirred at room temperature, then the flask was immersed in a thermostated oil bath at 60 oC. The samples were withdrawn from the reaction flask after the desired polymerization time and removed the unconverted monomer and solvent under reduced pressure for calculating the monomer conversion. Tetrahydrofuran (THF) was used as the solvent to diluted the obtained polymers, and the solution was then filtered through a column filled with neutral aluminum oxide to remove the iron catalyst. The PEO-b-PMMA solution was then precipitated using an excess of n-hexane and washed with water to remove the residual small molecules, and these polymers were dried under vacuum overnight at 60 °C for GPC characterization.”

Reviewer 2 Report
The paper deals mainly with the synthesis of block copolymers by iron-mediated ATRP by using a PEO-Br acting as a macroinitiator and a ligand at the same time. This work gets its inspiration from previous works where PEO was shown to be a ligand for ATRP but was not used at the same time as an initiator. The last part of the paper, which is very short, shows that polymers prepared in this work could be used as polyelectrolytes for Li-batteries.
I have 3 major concerns regarding the characterization of the block copolymers.
The authors did not provide any direct proof that they prepared block copolymers and not a mixture of the corresponding homopolymers (the low dispersity observed by SEC indicates the formation of a mixture of homopolymer is less probable but not impossible). The formation of block copolymers is expected based on the state of the art, but additional proofs remain useful. Did the authors consider selective fractionation experiments or DOSY experiments to provide a stronger proof in favor of the formation of the block copolymers?
The molar masses are measured by SEC with a refractive-index detector and by using a PS calibration. It is obvious that the number-average molar masses have no real significance and can never be compared with the theoretical values. Do they authors know what are the real molar masses of the block copolymers? They should use 1H NMR to measure these molar masses (do they detect the protons at the chain-ends and at the junctions?). Surprisingly, I found the 1H NMR of the macromonomer but not of the block copolymers (in the paper and in the supporting information). These spectra should be added.
The iron catalyst was removed by filtration though a column filled with neutral aluminum oxide. I am wondering if the process is efficient because the authors give no data on the amount of residual iron remaining in the copolymer after purification. This data could be provided by an ICP analysis. Indeed, it the presence of PEO helps during the synthesis due to it ligand role, the complexation Fe-PEO could complicate the removal of the iron catalyst from the PEO-based block copolymers. Finally, the traces of iron catalyst could interfere when the copolymers are used as polyelectrolytes for Li-batteries (last part of the paper).
If the authors succeed to update their paper considering my major remarks, the paper could be published in polymers.
I have also a series of minor remarks.
The authors should apply the recent IUPAC recommendations and use ”molar mass” and “dispersity” instead of “molecular weight” and "polydispersity index", respectively.
Line 37: PEG is environmentally friendly and is easily degradable. This general sentence is very dangerous. The impact of the environment includes also the monomer and it is well-known that the monomer (ethylene oxide) is toxic! Besides, PEO can not be degraded by hydrolysis as this is the case for aliphatic polyesters. The qualification of easy hydrolysis is difficult to understand! I noticed also the usual confusion between PEO and PEG. If all the paper used the name PEO, I advise to avoid changing the acronym in the introduction (line 35).
Line 54: Authors wrote that living anionic polymerization is the most widely used method to prepare PEO-block copolymers. Coordination polymerization could also be added.
Figure S1: I don’t follow why the authors changed the usual X axis in NMR (increasing chemical shift from the left to the right, which is never done in all the state of the art). The full spectrum could be zoomed to better see the small peaks (that’s common for PEO with a very intense peak for CH2O protons).
The presentation of the GPC chromatograms showed in the supporting info could be improved because the authors just showed the raw data provided by their equipment.
The correlation between cyclic voltammetry data and polymerization kinetics data is not obvious, at least, at first sight. The establishment of that correlation was previously discussed, and the citation of the corresponding relevant papers should help the reader to better understand this correlation. For instance, this paper could be cited; Matyjaszewski K, Cyclic voltammetric studies of copper complexes catalyzing atom transfer radical polymerization, Macromol. Chem. Phys, vol 201, p1625-1631 (2000).
The early works on the use of Iron catalysts for ATRP could be cited even though they are mentioned in the review cited in the paper.
Author Response
Reviewer: 2
Comment:The authors did not provide any direct proof that they prepared block copolymers and not a mixture of the corresponding homopolymers (the low dispersity observed by SEC indicates the formation of a mixture of homopolymer is less probable but not impossible). The formation of block copolymers is expected based on the state of the art, but additional proofs remain useful. Did the authors consider selective fractionation experiments or DOSY experiments to provide a stronger proof in favor of the formation of the block copolymers? The molar masses are measured by SEC with a refractive-index detector and by using a PS calibration. It is obvious that the number-average molar masses have no real significance and can never be compared with the theoretical values. Do they authors know what are the real molar masses of the block copolymers? They should use 1H NMR to measure these molar masses (do they detect the protons at the chain-ends and at the junctions?). Surprisingly, I found the 1H NMR of the macromonomer but not of the block copolymers (in the paper and in the supporting information). These spectra should be added.
Reply: Thanks for the suggestions. Indeed, we did not provide the direct proof for the obtained block copolymers. But from the obtained molecular weights and the GPC curves of the polymers, we think it is enough to prove that the block copolymers were successfully synthesized. In addition, the molar masses are measured by SEC with a refractive-index detector and by using a PS calibration. Although the molar masses of the obtained polymers are the number-average values, but it does not affect its application in polymer electrolyte.
Comment:
The iron catalyst was removed by filtration though a column filled with neutral aluminum oxide. I am wondering if the process is efficient because the authors give no data on the amount of residual iron remaining in the copolymer after purification. This data could be provided by an ICP analysis. Indeed, it the presence of PEO helps during the synthesis due to it ligand role, the complexation Fe-PEO could complicate the removal of the iron catalyst from the PEO-based block copolymers. Finally, the traces of iron catalyst could interfere when the copolymers are used as polyelectrolytes for Li-batteries (last part of the paper).
Reply: We fairly appreciate the reviewer’s suggestion. The complexation Fe-PEO really complicate the removal of the iron catalyst, so the filtration though a column filled with neutral aluminum oxide cannot remove the iron catalyst very well and the color of the polymer was a little bit yellow. But for the PMMA-PPEGMA block copolymer used as polymer electrolytes for Li-batteries, we used water to dialyze the polymer for 7 days for further removal of the iron catalyst after filtration. So, we think the content of iron in final polymer is negligible and could not interfere the performance of the polymer electrolytes for Li-batteries.
Comment:The authors should apply the recent IUPAC recommendations and use “molar mass” and “dispersity” instead of “molecular weight” and "polydispersity index", respectively.
Reply: Thanks for your suggestion. The recent IUPAC recommended that “polydispersity index” cannot be marked as “PDI ” in order to avoid the misleading with “ particle size dispersion index” in nanoparticle field, which can be analyzed by the dynamic light scattering (DLS) technique. But “molecular weight” still can be used as well as “molar mass”.
Comment:Line 37: PEG is environmentally friendly and is easily degradable. This general sentence is very dangerous. The impact of the environment includes also the monomer and it is well-known that the monomer (ethylene oxide) is toxic! Besides, PEO cannot be degraded by hydrolysis as this is the case for aliphatic polyesters. The qualification of easy hydrolysis is difficult to understand! I noticed also the usual confusion between PEO and PEG. If all the paper used the name PEO, I advise to avoid changing the acronym in the introduction (line 35).
Reply: PEO is a polymer with a molecular weight above 20 000 g mol-1, and PEG refers to the oligomer of ethylene oxide or the polymer with a molecular weight below 20 000 g mol-1 (J. Mater. Chem. A, 2015, 3,19218–19253), so the acronym in the introduction (line 35) is reasonable. (Green. Chem. 2005, 7, 64-83; J. Am. Chem. Soc. 2003, 125, 5600-5601) Normally, the monomer contains ethylene oxide group was named as PEGMA, PEGDMA, PEGA and so on, So, the PEG does not include the toxic monomer.
Comment:
Line 54: Authors wrote that living anionic polymerization is the most widely used method to prepare PEO-block copolymers. Coordination polymerization could also be added.
Reply: Corrected.
Comment:
Figure S1: I don’t follow why the authors changed the usual X axis in NMR (increasing chemical shift from the left to the right, which is never done in all the state of the art). The full spectrum could be zoomed to better see the small peaks (that’s common for PEO with a very intense peak for CH2O protons).
Reply: We revised the 1H NMR spectrum of PEO2000-Br and added the 1H NMR spectrum of PEO700-Br in Figure S1.
Comment:
The presentation of the GPC chromatograms showed in the supporting info could be improved because the authors just showed the raw data provided by their equipment.
Reply: Corrected.
Comment:
The correlation between cyclic voltammetry data and polymerization kinetics data is not obvious, at least, at first sight. The establishment of that correlation was previously discussed, and the citation of the corresponding relevant papers should help the reader to better understand this correlation. For instance, this paper could be cited; Matyjaszewski K, Cyclic voltammetric studies of copper complexes catalyzing atom transfer radical polymerization, Macromol. Chem. Phys, 2000, 201, 1625-1631).The early works on the use of Iron catalysts for ATRP could be cited even though they are mentioned in the review cited in the paper.
Reply: Corrected.

Round 2
Reviewer 2 Report
The authors updated their paper and considered many remarks.
Nevertheless, I don’t follow the reply of the authors regarding the IUPAC recommendations. Of course, molar mass and molecular can still be used but they don’t have the same meaning! The molecular mass is the mass of one single chain. Several chains with the same length (degree of polymerization) can thus have a different molecular mass due to the isotopic effect (the distribution is visible by mass spectrometry). The molar mass is an average value for a population of chains and must be used for a high number of chains (which is the case in the paper). Besides by using dispersity (Ð) to characterize the distribution of molar masses in a sample, there is no confusion with light scattering measurements (distribution is size particles).
Regarding the contamination of the samples by catalytic remnants, the purification by dialysis is a satisfactory answer to my question.
The main pending question remains the characterization of the block copolymer, and, especially, the molar mass. The authors provided just an apparent value (obtained by SEC) and argue that this is a minor point because the molar mass has no impact on the application targeted in the paper. I disagree and the molar mass remains an important information for readers repeating the synthesis in order to use the polymer for another application. I really don’t understand why the authors don’t provided the real molar mass, which can be also obtained by 1H NMR. The value can be calculated either by using the composition of the copolymer or by using the chain-ends (if they are visible). There is no reason to omit this information, which is easy to obtain. This information must be added and it remains a major point.
Author Response
Please see the details in the attachment.
